# Linking protein structural and functional change to mutation using amino acid networks

**Cristina Sotomayor-Vivas[1], Enrique Hernández-Lemus[1,2], Rodrigo Dorantes-Gilardi[3]***

**1** Department of Computational Genomics, National Institute of Genomic Medicine, Mexico City, Mexico,
**2** Center for Complexity Sciences (C3), Universidad Nacional Autónoma de México, Mexico City, Mexico,
**3** Center for Complex Network Research and Department of Physics, Northeastern University, Boston, MA, United States of America

* r.dorantesgilardi@northeastern.edu

## Abstract

The function of a protein is strongly dependent on its structure. During evolution, proteins acquire new functions through mutations in the amino-acid sequence. Given the advance in deep mutational scanning, recent findings have found functional change to be position dependent, notwithstanding the chemical properties of mutant and mutated amino acids. This could indicate that structural properties of a given position are potentially responsible for the functional relevance of a mutation. Here, we looked at the relation between structure and function of positions using five proteins with experimental data of functional change available. In order to measure structural change, we modeled mutated proteins via amino-acid networks and quantified the perturbation of each mutation. We found that structural change is position dependent, and strongly related to functional change. Strong changes in protein structure correlate with functional loss, and positions with functional gain due to mutations tend to be structurally robust. Finally, we constructed a computational method to predict functionally sensitive positions to mutations using structural change that performs well on all five proteins with a mean precision of 74.7% and recall of 69.3% of all functional positions.

**Data Availability Statement:** Data and code to create the figures can be found at https://github.com/CrisSotomayor/perturbation-networks.

**Funding:** This work was supported by CONACYT (grant no. 285544/2016 Ciencia Básica, and grant

## Introduction

Proteins are complex biomolecules that have been subject to mutational dynamics for billions of years and whose tasks are essential for the maintenance, development, and survival of well-functioning cells. They start from a sequence of amino acids that folds into a three-dimensional (3D) structure that determines their function [1]. Understanding the underlying relations between sequence, structure, and function of a protein has been an active research topic in molecular biology for decades [2, 3].

Structure and function prediction from the amino acid sequence has been an open problem even prior to Anfinsen's discovery of the thermodynamic hypothesis, which states that, under normal conditions, the protein sequence is responsible for the native configuration of a protein

no. 2115 Fronteras de la Ciencia), as well as by federal funding from the National Institute of Genomic Medicine (Mexico). Additional support has been granted by the National Laboratory of Complexity Sciences (grant no. 232647/2014 CONACYT). EHL acknowledges additional support from the 2016 Marcos Moshinsky Fellowship in the Physical Sciences. The funders have no role in the design or development of this project.

**Competing interests:** EHL is an Academic Editor at PLoS ONE.

[1, 4]. In the last couple of decades, widely available datasets of protein 3D structures like the Protein Data Bank [5], machine learning methods such as deep learning [6], as well as high-throughput methods to quantify functional scores at massive scales [7–9], have brought us closer to understanding the interconnections between protein sequence, structure, and function.

In particular, with the advent of the *big data paradigm* there has been a renewed interest in the laws yielding structure and function from the one dimensional amino acid sequence [10]. Machine learning methods developed to predict residue-residue contacts in the 3D structure have recently shown a relation between residue proximity and coevolution measured by the covariance of positions in homologous protein sequences [10–14]. Coevolving positions have also been shown to be functionally sensitive to mutations using deep mutational scanning data [15, 16], reinforcing their prime role in protein structure and function.

The replacement of an amino acid in the sequence—a mutation—can have structural consequences on the resulting protein and thus has a potential effect on its function. In general, mutations occur naturally and have no effect on the protein function: this is called protein robustness [17, 18]. Protein adaptation or evolvability also requires that some mutations can change the protein's function [19, 20], indeed, a mutation can make the protein obtain a different function [21, 22]. Finally, a small set of mutations can leave the protein without the original function [23], either because of loss or adaptation, yielding protein fragility.

Experimental evidence on the interrelation between function, structure, and mutation has been shown before. For instance, via the analysis of missense mutations of the tumor suppressor p53, where mutations at the DNA-binding structural domain were found to produce functional loss more often [24]. Computational studies of the effects of *in silico* mutations in protein structure have shown that most positions are structurally robust independently from the chemical properties of the mutant residue [25], and sensitivity depends on their structural neighborhood [26]. *Functionally-wise*, experimental research has shown that functional change (fragility or adaptation) is, in general, exclusively dependent on the sequence position mutated and not on the amino acid or its mutants [16, 27].

In the case of a mutation, the fact that sequence positions seem to contain the necessary information for structure and protein fitness, raises the question of the relation between functionally and structurally sensitive positions. Although deep mutational scanning has brought results in many areas of molecular biology [8], the availability of its data is not yet ubiquitous and it also has been often created for the analysis of epistatic effects and thus not including single mutations. This brings the additional question of whether a relation between structure and function can be observed by alternative, cheaper methods. Specifically, given a protein, can we obtain the functional relevance of its sequence positions by looking at its 3D structure?

Network science has been successfully used in biology to model a variety of systems including co-expression networks [28–31], metabolic pathways [19, 32], protein-protein interactions [33–36], detection of protein function [37], and protein structure [38–40]. Amino acid networks, where amino acids are represented by nodes that are connected if they are within a distance threshold, have been used to model protein structure [41, 42] and study the effects of mutation on structural fitness [25, 26]. A great advantage of computing structural change under this framework is the availability of more than 144,000 structural protein models based on their 3D atomic coordinates in the Protein Data Bank [43].

Here, we propose to use this methodology to study the relation between change in protein structure and function by considering five proteins for which deep mutational scanning data is available [16, 44–47]. For these proteins, the functional change resulting from a mutation has been quantified for all amino acid substitutions, in most sequence positions. We obtained

corresponding structural change data *in silico* using the perturbation network of a mutation obtained by comparing the 3D structure of the original protein and that of its mutation.

We found that structurally sensitive positions (SSPs) are not only position dependent but are also strongly correlated to functionally sensitive positions (FSPs) in all 5 proteins. Moreover, prediction of FSPs using SSPs yields a mean precision of 74.7% and recall of 69.3% across all five proteins. Moreover, the area under the receiver operating characteristic (ROC) curve, a quantity often used to assess the quality of the prediction, has a mean value of 0.83 ± 0.04, showing a clear relevance of positions' structure in functional fragility due to mutations.

To measure structural change, we considered three different topological measures of the perturbation network, namely its size (in nodes), its number of edges, and its weighted sum. In practice, the size of the perturbation network represents the number of amino acids affected by the mutation; its edges, in turn, represent the structural contacts between amino acids changed, and its sum of weights is the number of atomic pairs that either moved closer or further apart of a chosen distance threshold. We show that mean structural change of sequence positions accounted by each measure is correlated to experimentally-obtained functional change. However, aggregating the perturbation measures increases the correlation between functional and structural disruption. This relation was found for amino acid networks defined by 71 different atomic distance thresholds in the range of 3–10 Ångstroms (Å).

Comparing the scores obtained for predictions using a distance threshold of 9 Å with the scores obtained from all other thresholds in the 4–10 Å range, we observed that predictions using a 9 Å threshold achieve similar or better scores than all other thresholds. This is true across the five proteins studied and using all perturbation measures. We suggest that 9 Å is indeed a good choice of threshold for obtaining accurate predictions of FSPs independently from protein size.

Finally, the complement of the SSPs, the set of structurally robust positions (SRPs), correlates well with top 40% of positions with weaker functional loss (or with a gain in function). Within those positions many have a functional change close to zero, suggesting a relation between structural and functional robustness.

## Results and discussion

The relationship between structural and functional change studied here is based on the comparison between the perturbation network of mutations and their corresponding experimentally obtained functional change in five proteins. We combined three network-based measures representing structural change to ultimately be able to predict positions sensitive to mutations. Below is a summary of the results found:

- Structural sensitivity (or robustness) to mutations is position dependent.

- Significant correlations show that there is a relationship between protein structural and functional change due to mutations.

- Predictions for functionally sensitive positions based on individual network measures—nodes, edges or weight—achieve considerable scores. Aggregating multiple network measures to obtain predictions improves the precision.

- Stronger structural perturbation is related to stronger functional change.

- The use of network parameters allows us to design predictions maximizing different values, be precision, recall, or both simultaneously.

- A relationship between robust positions to mutations and those that have small functional change can also be observed.

## Distance thresholds

**Correlation between structure and function.** Weighted amino acid networks as we have constructed here are usually defined for distance thresholds between 5 Å and 8 Å, depending on the intended chemical interactions to capture [38]. Threshold distances for atom-atom interactions usually vary between 4.5 Å [48] and 5 Å [41, 49]. In general, the edges of amino acid networks are supposed to be at least loosely based on the underlying chemical interactions of the protein. Here, we took a different approach: we did not aim to model the biological interactions between amino acids, but their structural neighborhoods, spanning much larger distances than those chemically feasible [38].

Given a mutation, the perturbation network resulting from the comparison of a mutated structure to the original three-dimensional (3D) conformation quantifies the structural change of the mutation. Four parameters of the perturbation network were considered as perturbation measures, namely, its number of nodes, its number of edges and their weight sum, and its diameter (Methods).

To identify the best distance threshold to use, we first calculated Spearman correlation values between functional change of sequence positions and their perturbation-network parameters. For each protein and each parameter, we compared the mean functional value and the mean perturbation measure score per sequence position. Higher perturbation scores resulted from 3D structures farther away from the original, hence possibly more likely to have a disrupted function. This would be reflected by stronger negative correlations, relating higher perturbation scores with lower functional scores. For simplicity, we set all correlations to absolute values.

We found consistent results between the five proteins studied when comparing each perturbation measure to functional change (Fig 1). Mean and standard deviation Spearman correlation ($\rho$) for measure nodes were −0.56 ± 0.12, for edges −0.53 ± 0.1, for weight −0.51 ± 0.1, and for diameter −0.3 ± 0.11. For most measures we found statistically significant correlations between structural and functional change. For measures nodes, edges, and weight the correlations were significant (mean $p$-value = $3.6 \times 10^{-4} \pm 6.2 \times 10^{-3}$), however that was not the case for the diameter of the perturbation network (mean $p$-value = $1.6 \times 10^{-2} \pm 5.3 \times 10^{-2}$).

For measures nodes, edges, and weight, we found that correlations increased steadily for thresholds between 3 and 4 Å, showing a slight peak around 3.8 Å, and then stabilized around 4 Å for correlations between 0.3 and 0.65. In the case of the measure 'diameter', correlations

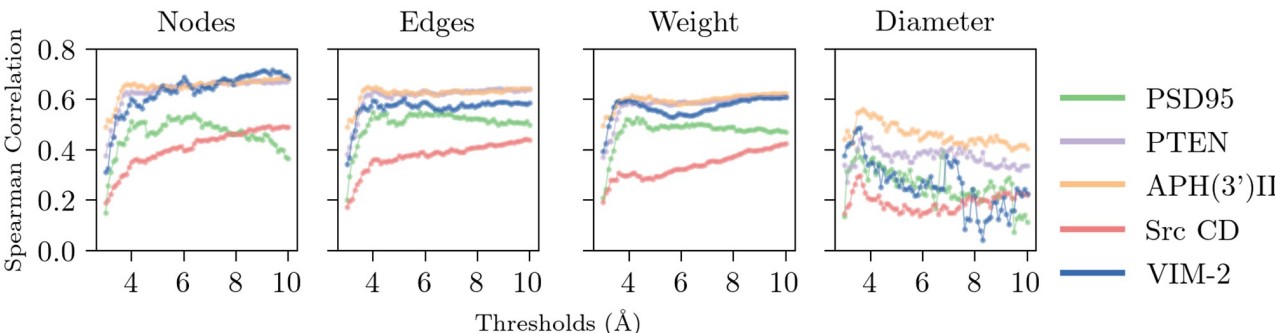

**Fig 1. Spearman correlation between positions' mean structural and functional scores by protein, perturbation measure, and distance threshold.**

peaked between 3.5–3.8 Å for all five proteins and then decreased for higher distance thresholds.

Relations between structure and function shown here suggest that protein structure can be studied with much higher distance cutoffs ($\sim$9 Å). By arbitrarily ignoring chemical-based interactions we were able to better account for the structural change around a mutated position, suggesting that studies exclusively looking at the protein structure may benefit from including higher distance thresholds.

**Prediction of functionally sensitive positions.** We also analyzed predictions considering exclusively individual measures, that is, given a measure, we set a perturbation cutoff and selected all positions that had at least one mutation above the cutoff (S1 Fig). We considered cutoff 1.5, representing 1.5 standard deviations above the mean, and looked at both precision and recall (Methods).

To compare the different perturbation measures, we took the predictions obtained from single measures and averaged scores over all thresholds and proteins for each measure. We found that the number of nodes had the highest mean precision (72.66%), weight had the highest mean recall (71.76%), and diameter had the lowest score in both cases (52.58% and 49.02%, for precision and recall, respectively).

With these correlations and predictions based on single measures, we saw that in most cases, we got more information from higher thresholds, reflected by higher correlations and precision scores. Since the diameter of perturbation networks had less predictable behavior compared to the other three measures, lower correlation scores, and lower scores when predicting based just on this measure, we will not include it when making predictions of functional positions. We believe that this measure is too sensitive as adding or removing a single edge could significantly change the maximal smallest path without significantly changing the network itself; its sensitivity to small threshold changes is can be seen in Fig 1. For nodes, edges, and weight, we considered the average scores between the 5 proteins and 3 measures, and found that precision is maximized at 9.3 Å, while recall is maximized at 8.4 Å, suggesting that an optimal threshold can be found in that range. Hereafter, we considered 9 Å as the representative threshold.

## Perturbation cutoffs and minimum counts

**Aggregating perturbation measures.** When selecting perturbation cutoffs and minimum counts—the cutoffs defining structurally unstable positions and the number of altered measures required for instability, respectively (Methods)—we started from the idea that stricter predictions, those arising from higher cutoffs and counts, reflected higher structural changes. In other words, we assumed that the more the structure of the protein was modified, the more likely it was that the function was disrupted. Hence, we expected stricter predictions to result in higher precision.

Testing different cutoffs and minimum counts confirmed this hypothesis, as well as the fact that more lenient predictions were more likely to have a higher recall, while sacrificing precision (S2 Fig, Fig 2, Table 1). In Table 1, we can see that the mean precision increased as the number of perturbation measures considered (minimum count) increased, while recall decreased. When we considered only one perturbation measure, we got a mean precision and recall of 65.96% and 82.58%, respectively. Inversely, when considering all three measures, we obtained a mean precision of 78.6% and recall of 51.47%. This shows that aggregated scores predict better than single scores when the aim is to obtain higher precision, suggesting that the three perturbation measures are relevant to account for structural change.

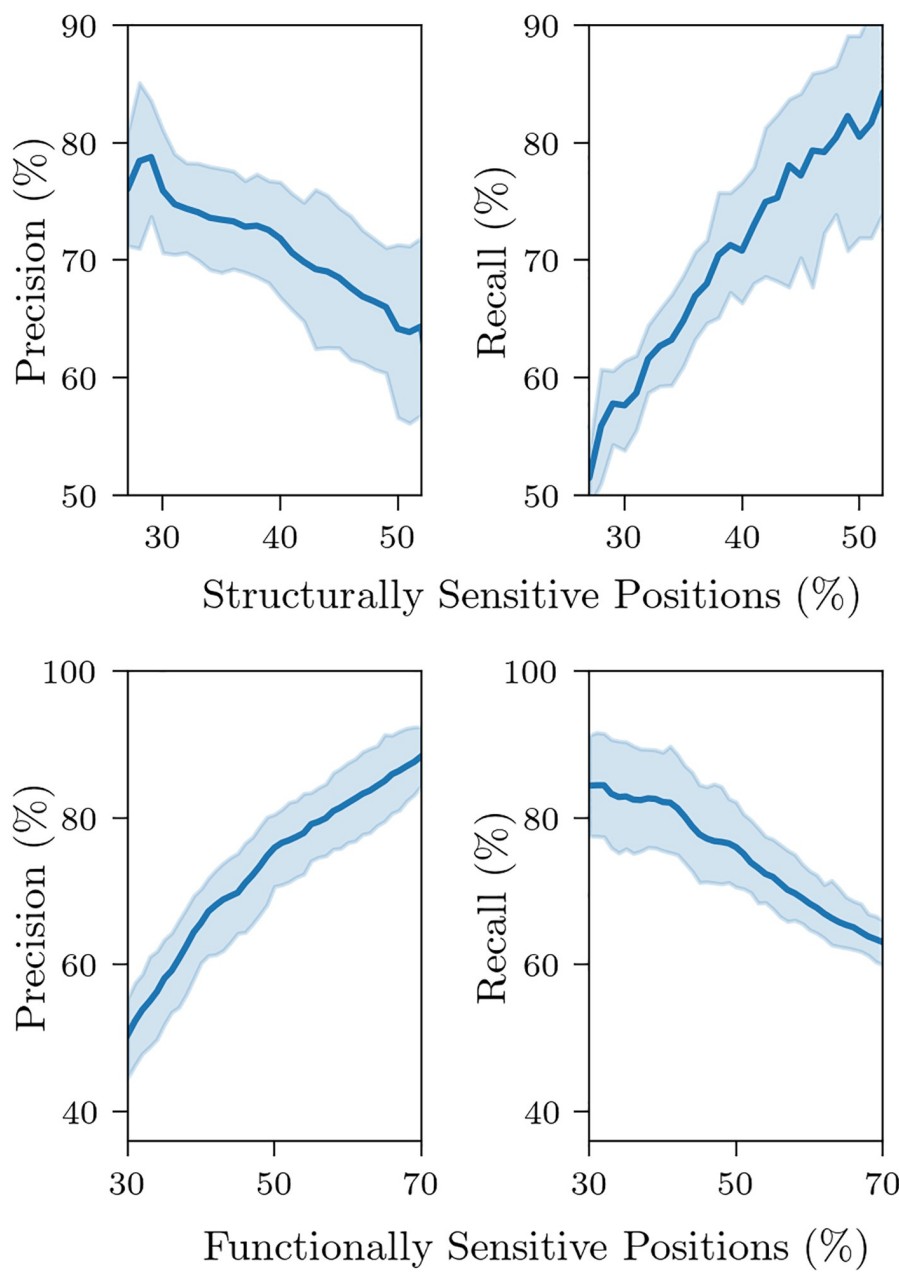

**Fig 2. Top, comparing precision and recall scores with functional percentage, leaving parameters fixed at (1,1,1) and minimum count of 2, varying functional percentage from 30 to 70%; bottom, comparing precision and recall scores with prediction percentage, leaving functional percentage fixed at 40%, minimum count fixed at 2, and varying the functional cutoffs from 1 to 2 to obtain different prediction percentages (percentages were rounded and missing values filled in through linear interpolation).** The line represents the mean over the proteins, while the shaded area represents 95% confidence interval.

**Percentages of FSPs and SSPs.** Precision and recall scores are also closely related with the percentage of functionally sensitive positions (FSPs), and structurally sensitive positions (SSPs, predicted positions), respectively. In Fig 2, we compared how changes in these percentages were reflected in the precision and recall scores. To obtain changes in the prediction percentage, we varied the cutoffs from 1 to 2, in intervals of 0.02, for a total of 51 cutoffs. As cutoffs

**Table 1. For all three minimum counts, we evaluated predictions with 51 different perturbation cutoffs ranging from 1 to 2 (same cutoff for all three measures), and calculated the mean over all cutoffs and proteins, obtaining a mean score for precision and recall for each minimum count.**

| Minimum count | Mean Precision (%) | Mean Recall (%) |
|---|---|---|
| 1 | 65.96 | 82.58 |
| 2 | 72.78 | 67.42 |
| 3 | 78.6 | 51.47 |

increased, structurally sensitive positions decreased, which was reflected in higher precision and lower recall, providing further evidence on the relationship between stricter measures and higher precision scores.

In the range between 18–30% of SSPs, we obtained at least 75% of precision. Larger percentages of SSP decreased precision in all proteins. Positions captured with stricter cutoffs and minimum counts had mutations with larger perturbation networks relative to mutations at other sequence positions. This may be due to some particularity in their 3D structural neighborhood, whose interconnections are sensitive to most mutations. In this sense, the more unique the 3D neighborhood of the position, the greater the mean structural change is to be expected. An example may be the active sites in enzymes, which usually take a different substructures from the rest of the protein, whether it be a pocket, a cleft, an oligomeric interface, or another 3D shape [50]. Indeed, mutations happening at or close to active sites tend to affect the protein activity either by enhancing it [51, 52], losing it or adapting it [53].

Similarly, positions within an allosteric path which conveys signals from the active site to a distant position are found to be co-evolving within protein families, which in turn tend to be functionally sensitive to mutations [54]. These positions could be subject to structural particularities in their close neighborhoods, relative to other positions, and thus having greater structural changes. A more thorough analysis of the structural perturbation and its relation to the neighborhoods of biologically relevant positions is needed in this regard.

In all of our predictions we considered perturbation cutoffs and minimum counts such that the percentage of structurally sensitive positions returned was informative, ranging from around 25% when maximizing precision to around 50% when maximizing recall. As a basis, we compared all of our predictions with the 40% of positions with lowest functional values, and we henceforth refer to them simply as functionally sensitive positions, or FSPs. Leaving all other values fixed (perturbation cutoffs, minimum count, and distance thresholds), increasing this percentage led to a better precision and lower recall, while decreasing this percentage had the opposite effect (Fig 2). We focused on 40% as a balance between obtaining more precise predictions and selecting positions with significant disruption in their function.

With distance cutoffs and functional percentage fixed, we focused on combinations of perturbation cutoffs and minimum counts to make different predictions. For all of them, we considered the precision, recall and improvement scores (Methods), the latter representing the ratio between obtained scores and expected scores from random predictions.

Since the correlations between the mean structural and functional scores by protein showed similar scores among the thresholds 4–10 Å for nodes, edges and weight scores (Fig 1), we calculated the precision, recall and improvement scores for each of the possible thresholds for the three measures, to compare them to the predictions we obtained using the threshold 9 Å for those measures and evaluate the choice of parameters (Fig 3).

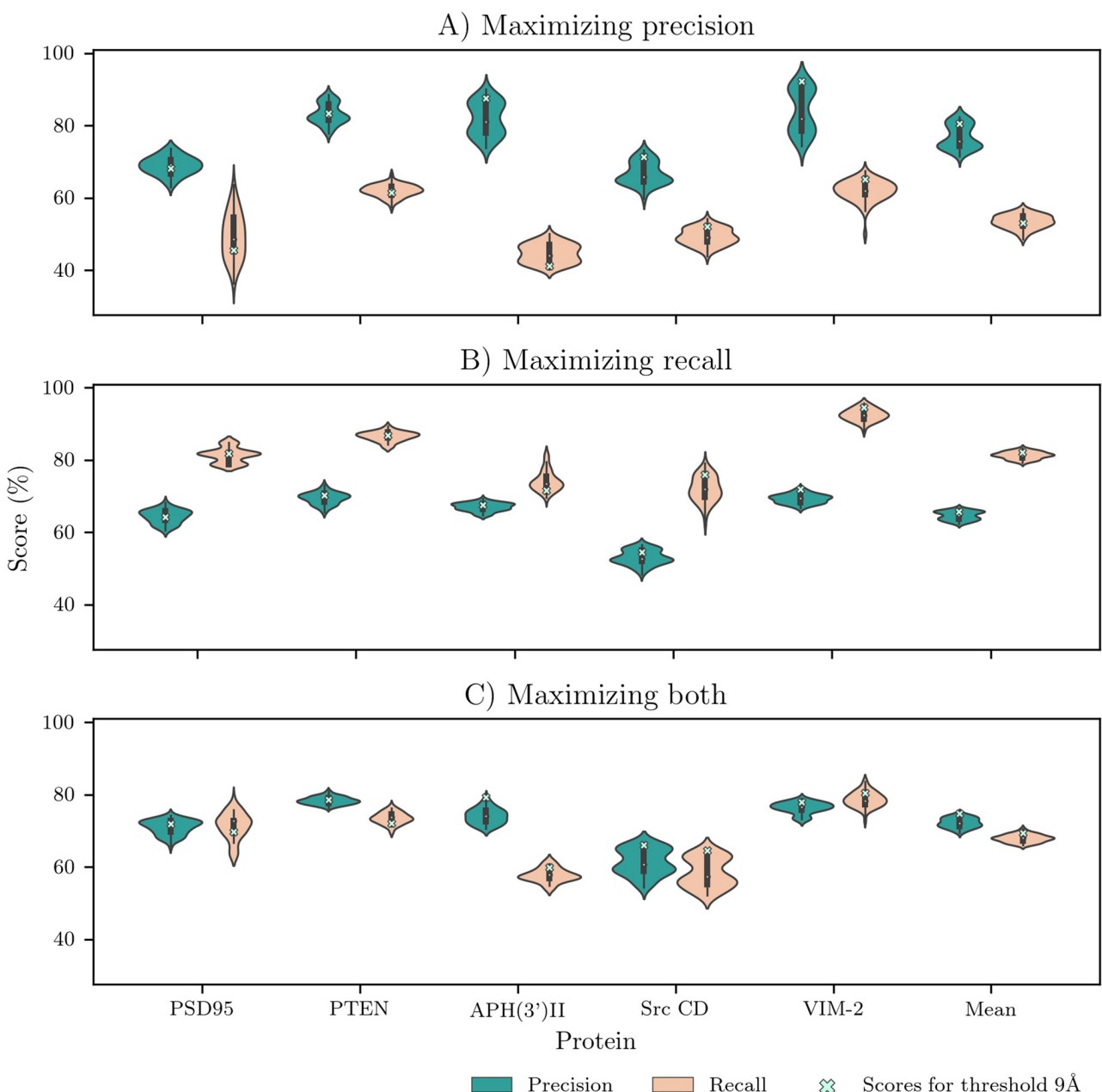

**Fig 3. Comparing the precision and recall obtained from varying the threshold for nodes, edges and weight scores from 4 Å to 10 Å, with threshold 9 Å, representing our predictions for FSPs, highlighted in red.** Different minimum count and cutoff vectors were used to A) Maximize precision, B) Maximize recall, and C) Maximize precision and recall.

## Protein structure-function relation

**Distance threshold of 9 Å.** Having established that stricter measures result in predictions with higher precision but lower recall and vice versa, we considered three sets of parameters to give predictions focusing on high precision, high recall, and an equilibrium between both. We refer to these predictions as maximizing precision, maximizing recall, and maximizing both.

In Fig 2, we varied cutoffs between 1 and 2, and looking at the prediction percentages, we obtained ranges of 38.7% to 63.2% for minimum count 1, 26.6% to 50% for minimum count 2,

and 15.1% to 39.1% for minimum count 3 (with lower values for cutoff 2, higher for 1). Based on this, and the known behavior of the parameters, we chose the minimum counts and cutoffs for predictions depending on the value to maximize and to keep informative prediction percentages: lower to maximize precision and higher to maximize recall.

First, to maximize precision, we selected stricter measures, considering a minimum count of three, as it had the highest mean precision, and a perturbation cutoff vector (1.5, 1.5, 1.5). This resulted in a mean precision of 80.5%, a mean recall of 53.1%, and a mean prediction percentage of 26.3% (Fig 3A), as well as an improvement by a factor of 2. In other words, using the three perturbation measures to account for structural change, we got a set of functionally sensitive positions with high precision.

In other studies, coevolving positions in protein families obtained using statistical coupling analysis, usually called protein sectors, have been found to form physically connected subnetworks of amino acids [15, 55]. Most of these positions, around 20% of all sequence positions, have been found to be sensitive to mutations. In particular the protein sector of the PSD$^{pdz3}$ protein, one of the proteins studied here, has been related to functional loss from single mutations [16]. Predicted positions maximizing precision, showing a similar percentage of the amino acid sequence, may also be related to protein sectors in other proteins but further researcher is needed in this direction.

Next, to maximize recall, we focused on more lenient measures. We considered the perturbation cutoff vector consisting of all ones, and a minimum count of 2. This minimum count showed more balanced results between precision and recall, while the perturbation cutoff vector (1, 1, 1) helped maintain a high recall. Comparing those predictions with the functionally sensitive positions, we found a mean recall of 82.2% over the 5 proteins studied, with a mean precision of 65.7%, while the mean percentage of predicted positions was 50% (Fig 3B). This resulted in an improvement of random predictions by a factor of 1.64.

A more general prediction, aiming to maximize precision and recall simultaneously, was achieved by once again using the perturbation cutoff vector (1.5, 1.5, 1.5), but with a minimum count of 2. This prediction resulted in a mean precision of 74.7%, a mean recall of 69.2%, and a mean prediction percentage of 37% (Fig 3C), as well as an improvement by a factor of 1.87. By predicting roughly the same number of positions as the number of FSPs, we believe this combination of parameters is a good general prediction if there are no functional values to compare to.

**Functional prediction from SSPs.** As we can see in Fig 4B for VIM-2 protein, in S3–S7 Figs for the other four proteins, and in Table 2, structural change due to mutation, similar to functional change, is position-dependent and independent from chemical properties of either the mutant amino acid or the one being replaced, supporting similar results from previous work [26]. This position dependence is also found in terms of functional change from mutations (Fig 4A, Table 2), suggesting that both structure and function relevance is determined by the position and not by the amino acid occupying it. Notably, structural measures with a higher position independence also showed higher correlations with functional change. This further supports the importance of the structural neighborhoods of positions disregarding chemical bonds to study protein structure.

To further evaluate our model, we obtained the receiver operating characteristic curve, plotting the True Positive Rate against the False Positive Rate. We fixed the threshold at 9 Å and varied the perturbation cutoff from 0.03, to 3. As we have seen, changing this value varies the recall, or True Positive Rate, with lower values corresponding to a higher recall. For a minimum count of two, evaluating predictions for sensitive positions resulted in a mean area under the curve of 0.83 ± 0.04 (Fig 5).

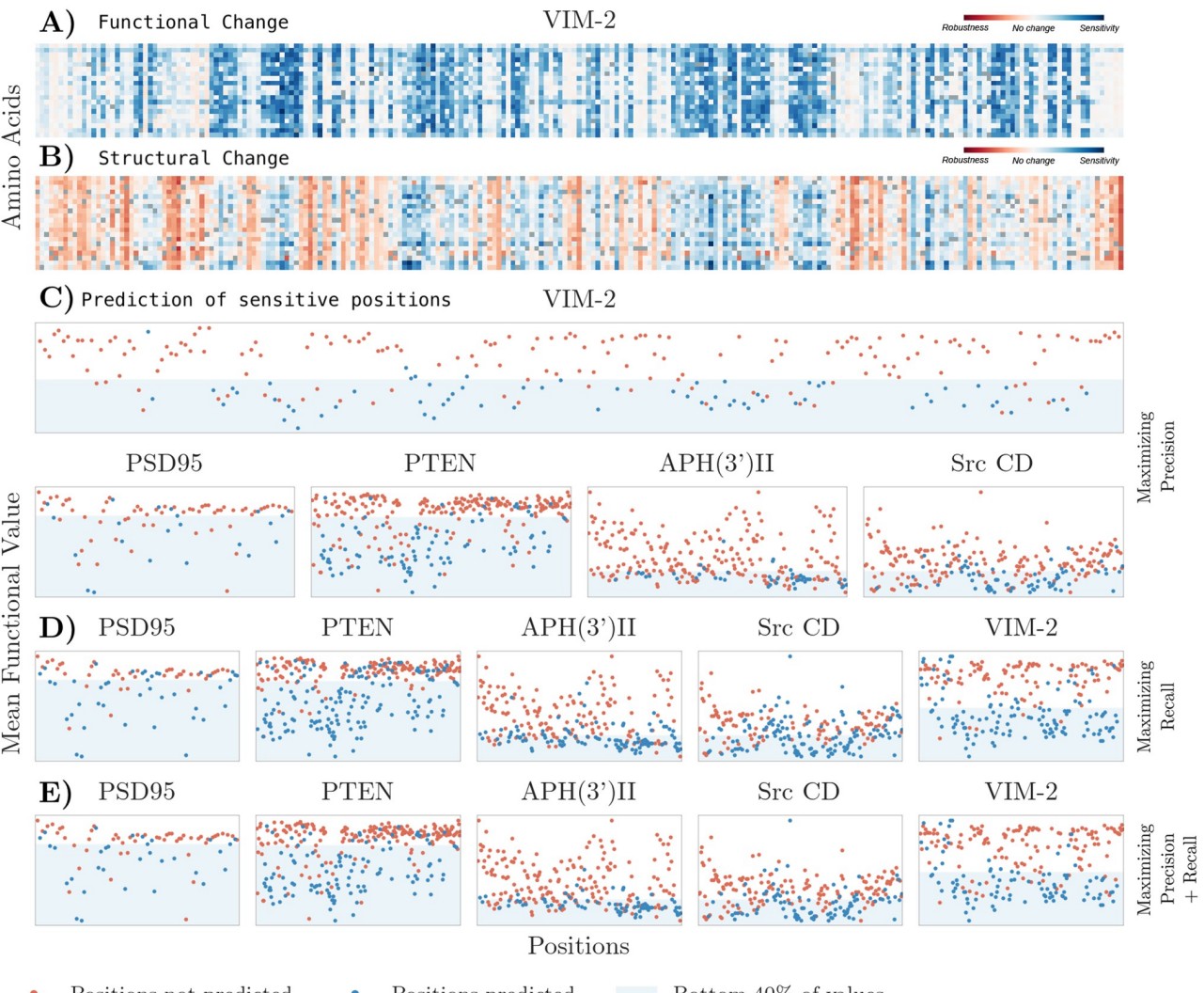

**Fig 4.** A) Experimentally obtained functional data from deep mutational scan of VIM-2 protein, with darker values representing higher functional disruption, specifically blue is loss of function while red represents gain of function [47]. B) Standardized data of the number of nodes perturbed by each mutation where each entry is the number of standard deviations from the mean of the distribution. The perturbation network was constructed using a threshold of 9 Å; blue represents highest structural perturbation, and red represents lowest. C) Predictions maximizing precision. X-axis has the sequence positions, Y-axis has the experimentally obtained mean functional value. Blue dots are SSPs—our predictions for FSPs—while shaded blue area contains the 40% of sequence positions with lowest functional scores representing strongest functional loss. Top row shows the functional values experimentally obtained for VIM-2 protein, bottom row the other four proteins studied. D) Predictions maximizing recall. E) Predictions maximizing both measures.

High precision to predict functional positions reinforces evidence about the relation between function and structure. Interestingly, on average more than 80% of highly SSPs (top 26%) tended to also be FSPs for all five proteins studied, yielding a precision that to our knowledge is not yet met in other non-experimental scenarios. Our framework could have applications in fields where a high precision in determining non yet known functional positions could be of significance, e.g. to inhibit the function of a target protein related to disease by a single mutation. Moreover, precision rates of around 70% were similar using perturbation networks and coevolving positions [16]. Given a protein, the advantage of our method is the lack

**Table 2. Considering structural and functional data, we looked at perturbation values per position, and considered the percentage of positive scores and negative scores, keeping the maximum of the two.** This presents a measure of consensus between the changes at each position, the higher values represent that most mutations result in the same effect (whether positive or negative values), independent of the mutant amino acid or the amino acid being replaced. We present averages and standard deviations over positions and proteins for the nodes, edges, and weight measures (structural data) and for the functional data.

| Measure | Positions sharing same sign (%) |
|---|---|
| Nodes | 87.9 ± 14.4 |
| Edges | 83.9 ± 15.8 |
| Weight | 77.7 ± 15.3 |
| Functional data | 81.4 ± 15 |

of need of its protein family to predict its functionally sensitive positions, thus enlarging the scope of proteins to be used.

## Structural and functional robustness

Our predictions so far have focused on identifying positions likely to be functionally sensitive (FSPs). Thus, by turning to the positions left out of a certain prediction, we could identify those more likely to be functionally robust (FRPs) to mutation. We compared these new predictions, obtained from the complement of different predictions for unstable positions, with the 40% of positions with highest mean functional scores (Fig 6). This percentile of positions represents those with a gain of function, or those with small functional changes resulting in scores similar to the WT amino acid at that position.

In this case, we considered the cutoff vector consisting of only ones to define the structurally sensitive positions. By considering the complement of these positions as structurally

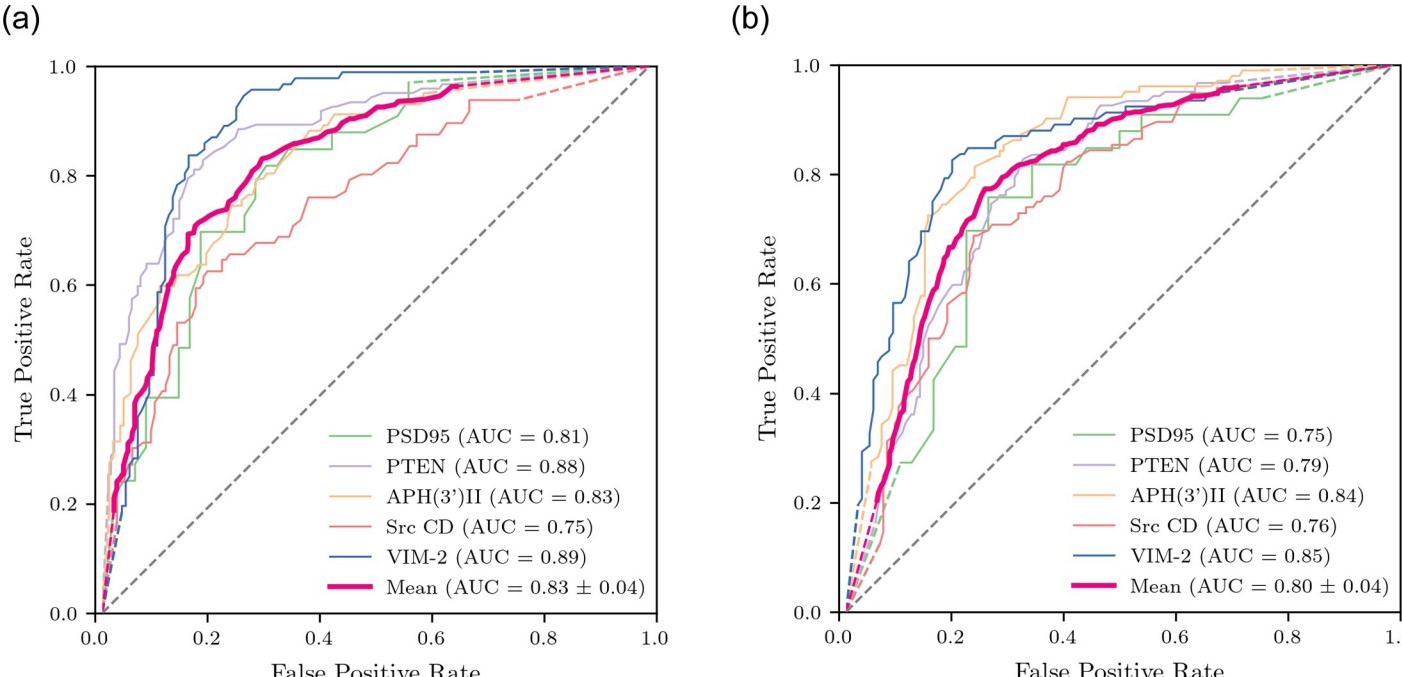

**Fig 5.** ROC curves for predictions of A) sensitive positions and B) robust positions are shown. ROC curves were obtained from varying cutoff-vector elements from 0.03 to 3, and fixing a minimum count of 2.

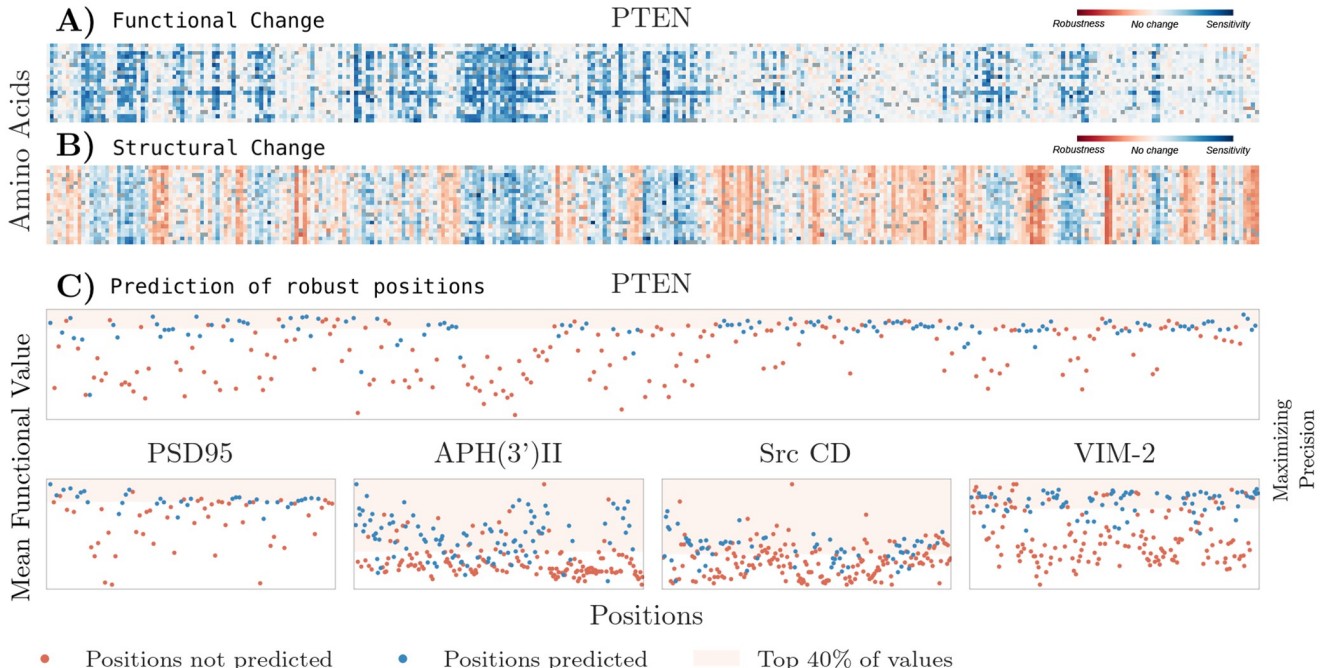

**Fig 6.** A) Functional data from deep mutational scan of PTEN protein, with lighter values representing smaller functional disruption, specifically blue is loss of function while white/red represents functional robustness to mutation [44]. B) Standardized data of the number of nodes perturbed by each mutation where each entry is the number of standard deviations from the mean of the distribution. The perturbation network was constructed using a threshold of 9 Å; blue represents highest structural perturbation, and red represents lowest. C) Predictions maximizing precision. X-axis has sequence positions, Y-axis has mean functional value. Blue dots are SRPs—our predictions for FRPs—while shaded red area contains the 40% of sequence positions with higher functional robustness. Top row shows PTEN protein, bottom row sows the other four proteins studied.

robust, we kept positions where all mutations had scores less than one standard deviation above the mean, those closer to the wild type. Since we considered the complement of sensitive positions, smaller minimum counts lead to stricter predictions for robust positions and vice versa.

We considered one prediction, with a minimum count of one, to showcase the relationship between structural and functional robustness. The minimum count of one guarantees that all mutations for predicted positions are less than one standard deviation above the mean for all three measures. This prediction had a mean precision of 70.4%, mean recall of 65% and a mean prediction percentage of 36.8%, which resulted in an improvement of 1.44. Once again we calculated the receiver operating characteristic curve, with threshold 9 Å. Predictions for robust positions with a minimum count of one resulted in a mean area under the curve of 0.80 ± 0.04 (Fig 5).

We also tested predictions for robustness analogously to those for sensitivity, selecting positions with at least one value below a threshold for a specific number of measures. This resulted in predictions with lower precision, between 50 and 60%, compared to 70.4% as presented above. This shows that, while a single 'bad' mutation can be telling of a sensitive position with high precision, the same cannot be said for 'good' mutations and robust positions. Instead, we find good predictions for robust positions when all mutations have scores closer to the mean, suggesting higher constraints for stability are required from the structural neighborhoods.

Protein evolvability depends on the ability of a protein to obtain a new function from a set of mutations (protein innovability), as well as in protein robustness (ability to withstand mutations) [17, 56]. Specifically, robustness is the ability of the protein to maintain both structure

and function in the case of mutations. The fact that over all five proteins, on average 70% of the top 30% structurally resilient positions were also within the most functionally robust may be a consequence of this property.

## Concluding remarks

We set out to explore the relationship between change in protein structure and function through the use of protein three-dimensional coordinates, *in silico* mutagenesis, and published deep mutational scanning datasets. We developed a method to predict functionally sensitive positions using structural data, and found a mean precision of 74.7% and a mean recall of 69.2% when comparing the predictions to functionally sensitive positions. By considering the complement of a set of predictions as structurally stable positions, we found a mean precision of 70.4% and a mean recall of 65% when comparing to the 40% of positions with highest functional values. Predicting randomly would lead to precision values close to the 40% of positions deemed functionally sensitive (or stable), and these predictions improve random predictions by factors of 1.87 and 1.44, respectively.

By changing the prediction parameters, we were able to obtain predictions with higher precision or recall, and we found a relationship between stricter parameters for structural sensitivity, requiring a bigger effect in the perturbation network, and more precise predictions. When predicting stable positions, more lenient parameters for sensitivity translate to stricter requirements for stability, and the same effect on precision was obtained. This supported a close relationship between structural and functional change in a protein. On the other hand, more lenient predictions for structural sensitivity lead to a greater recall, which relates to the greater percentage of positions included in the predictions. Our predictions maximizing precision improve that value by a factor of 2 for sensitive positions, compared to random predictions.

The method described can be used to predict sensitive positions in a protein without resorting to experimental methods, and it can be used as a standalone or in combination with other variant effect predictors [57], with the advantage that only the three-dimensional coordinate file is required as well as its *in silico* mutations. By knowing how the choice of parameters relates to the precision and recall in the proteins studied, we can estimate the probability of certain positions being functionally sensitive, and combine predictions to obtain positions most likely to be functionally sensitive, and predictions likely to encompass most functionally sensitive positions.

The predictions we considered and their respective scores show that it is harder to predict which positions are likely to show functional values above zero, showing gain of function, or close to zero, showing little or no functional change. However, we were able to observe a clear relationship between structural and functional robustness by looking at their correlation and their mutual position dependence.

The present approach may result particularly relevant in the design of protein structures via directed mutagenesis methods [58, 59]. Protein structure-based drug design [60, 61] either with pharmaceutical and biotechnological applications or even in terms of disease modeling, relevant in the context of, for instance, the COVID-19 pandemic [62, 63]. Also of contemporary relevance are the potential applications of our approach in the context of protein structural and functional prediction of CRISPR-Cas9 modifications [64–68]. The common scenario is that CRISPR-Cas9 genome editing allow us to determine gene sequences via highly specific modifications. Less clear are, however, the potential impact that such gene specific changes may bring to protein structure and function. In view of the plethora of applications of CRISPR-Cas9 genome editing in health, agriculture and biotechnology, it will become useful to have tools to predict, although still approximately, such effects.

A potential continuation of the work presented here is the use of machine learning models for classification prediction of functional positions trained on structural data from perturbation networks. In the same direction, if the structural data could be obtained exclusively from the 3D atomic coordinates, using network parameters local to the position, these models would not need further mutagenesis software. The need for this software represents the major weakness of our approach, as it require the availability and know-how of a third party software, a trade-off for not requiring additional data beyond the atomic coordinates of the protein. However, the computational time required was less or similar than other state-of-the-art methods showing good agreement with our predictions, such as DynaMut [73] (see Methods). The position dependence of structural change should incite further research on the identification of atypical neighborhoods in the structural vicinity of a position, and their relation with functional sensitivity to mutations.

## Methods

### Code availability

All the code used in this work is publicly-available at https://github.com/CrisSotomayor/perturbation-networks.

### Protein selection

We selected five proteins with published deep mutational scanning data and corresponding three-dimensional coordinates available in the Protein Data Bank [5], focusing on enzymes with substrate binding assays. The proteins selected were PSD95$^{pdz3}$ (PDB: 1BE9) [16], phosphatase and tensin homolog (PTEN) (PDB: 1D5R) [44], APH(3')II (PDB: 1ND4) [45], Src kinase catalytic domain (Src CD, PDB: 3DQW) [46], and VIM-2 metallo-$\beta$-lactamase (PDB: 4BZ3) [47].

### Functional change

We used the deep mutational scanning data to obtain functional scores for individual mutations. We considered the mean functional change at each position: the average score for all mutations at a particular position for which scores are available. Using these values and the percentage of positions we want to consider, we define functionally sensitive positions (FSPs) by sorting positions and selecting said percentage of positions with the greatest loss of function: the lowest mean functional change. Similarly, functionally robust positions (FRPs) are defined by selecting a percentage of positions with the weakest loss of function: the highest mean functional values. These values translate to positions with positive mean values or values close to zero. Throughout this paper we will consider 40% of positions for both FSPs and FRPs.

### Amino acid networks

Given the three-dimensional atomic coordinates of a protein and a distance threshold $t$, an amino acid network $G(t)$ is a network where nodes correspond to sequence positions and an edge between two nodes exists if there is a pair of atoms, one in each amino acid, at distance less than $t$. Moreover, each edge in the network has a weight corresponding to the number of atomic pairs at distance less than $t$ between the two nodes.

The construction of the networks was done in the `Python` programming language and implemented in a library called `Biographs` [69] based on the popular libraries `NetworkX` [70] and `Biopython` [71].

## Perturbation networks

Corresponding structural change data is obtained by first producing the same mutations *in silico* for each protein. Then, the resulting 3D structure of each mutation is modeled with an amino acid network and compared to the network of the wild-type 3D structure. The structural change of the mutation is represented by the topological difference of the two networks and called the perturbation network of the mutation, which accounts for the structural change of the protein. In this model, each topological measure of the perturbation network quantifies an effect of the mutation on a different structural property of the protein, and can be used to identify structurally sensitive positions to mutations. The full details of how we constructed the perturbation networks are described below.

For each protein, we performed *in silico* mutagenesis using the algorithm `FoldX 5.0` [72]. We mutated every position with corresponding functional data to the other nineteen amino acids. For each mutation, `FoldX` yields a three-dimensional structure that was used to construct the mutation's amino acid network. For each protein, its corresponding wild type amino acid network is obtained from the original structural file (PDB file). Using multithreading with 30 cores, computing the structures for all the point mutations took less than 24 hours per protein.

The perturbation network of a specific mutation is obtained by comparing wild type and mutation networks [42]. Given a distance threshold *t*, let *A* and *B* denote the adjacency matrices for the wild type and mutation networks *G*(*t*) and *M*(*t*), respectively. Let matrix *C* denote the absolute difference of the matrices *A* and *B*:

$$C = |A - B|.$$

After removing all the rows and columns containing only zeros from matrix *C*, we obtain the adjacency matrix that defines the perturbation network *P*(*t*) (Fig 7). We consider four topological attributes of *P*(*t*), namely its size (referred here as 'nodes'), number of edges ('edges'), total sum of weights ('weight'), and its diameter (maximal smallest path, called here 'diameter'). However, the diameter was ultimately not considered for making predictions.

## Structural change

For proteins with multiple identical chains, we obtained the perturbation networks for all mutations of all chains, and then calculated the average of each mutation over the different chains to obtain a single score per position. In order to capture a broad range of atomic distances, we constructed all networks using 71 different thresholds between 3 Å and 10 Å, where consecutive thresholds are spaced by a 0.1 Å step.

The four perturbation measures have different scales, and vary in magnitude according to the distance threshold, so we standardized the data to make comparisons between the different measures. We considered four data arrays per protein and threshold, one for each measure, containing the corresponding scores of every possible mutation (4 measures × 5 proteins × 71 thresholds = 1420 data arrays). We removed the null scores resulting from mutation to the same wild type amino acid to preserve the range of values obtained from non-synonymous mutations, and then standardized each array. A visual comparison of the standardized perturbation network data and the functional data is shown in Fig 4.

Given the standardized data, every mutation in every protein has four scores called nodes, edges, weight, and diameter, respectively. We refer to these four scores as the perturbation scores.

A)

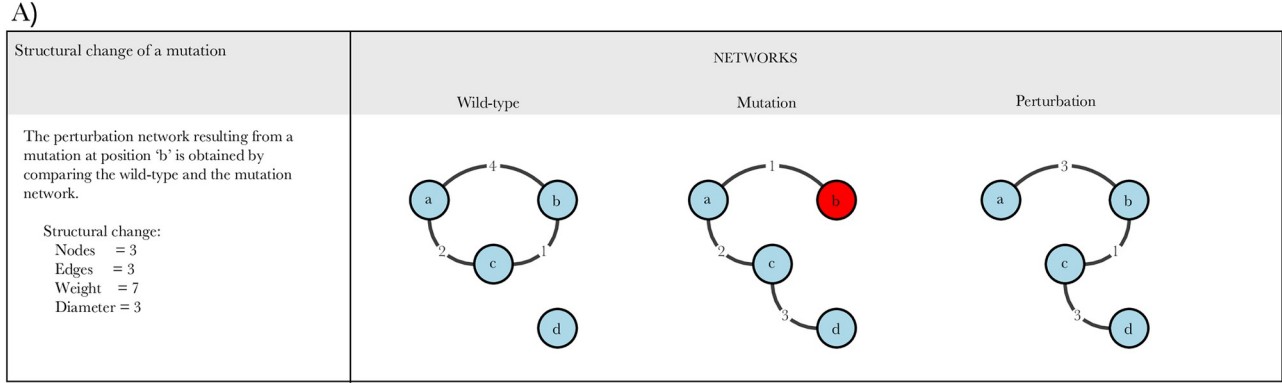

B)

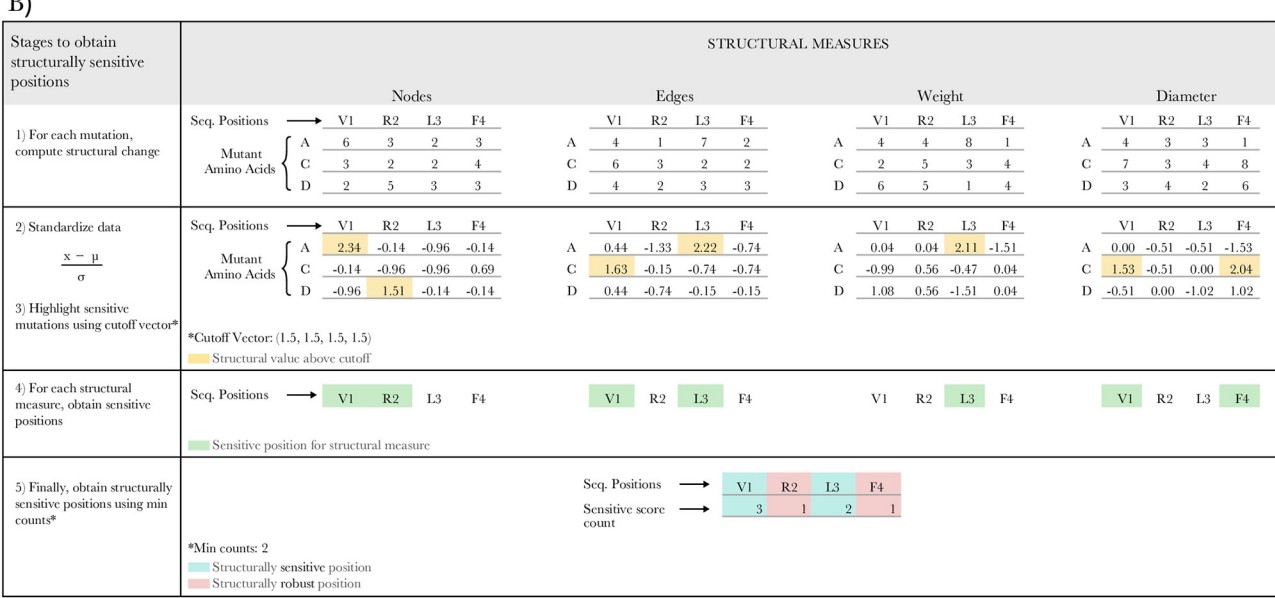

**Fig 7. Example of an amino-acid network _G_ with three nodes and three edges.** The network _M_ represents a mutation in node _b_, resulting in nodes _a_ and _b_ losing three pairs of atoms, and nodes _c_ and _b_ losing one edge. The network _P_ is the perturbation network of the mutation. In this example, _P_ has 3 nodes, 2 edges, weight 4, and diameter 2.

## Data standardization

In order to make all structural measures comparable across all mutations, the value of the perturbation of each mutation is measured in standard deviations from the mean, that is, the mean of the perturbation of all mutations was subtracted to each value and divided by the standard deviation. The perturbation value of each of the structural measures considered (nodes, edges, and weight) is given by the absolute difference between the network obtained by the mutation and the original (Wild-type) network, thus negative values are closer to the original network's values while positive values show a stronger structural perturbation.

Functional data was not standardized. In order to define the functionally sensitive positions we used the bottom 40% of the positions in terms of mean functional change (smaller values). This was done as we found the data to have varying distributions and few positions above the standard deviation cutoffs due probably to independent experimentation and methodology of the measuring of the functional change of mutations.

## Defining structural sensitivity

Since each measure represents different changes in the perturbation network, and therefore on the protein structure, each provides a different way to identify structurally sensitive positions (SSPs). First, we say that a mutation is sensitive for a certain measure if its corresponding perturbation score is above a particular perturbation cutoff. Given the four perturbation scores, we consider a perturbation cutoff vector containing four values including the specific cutoffs for sensitivity in terms of nodes, edges, weight and diameter measures, respectively. Thus, modifying the values in this vector yields different structurally sensitive mutations. Each cutoff corresponds to the number of standard deviations from the mean taken in each distribution, e.g. a cutoff of 1.5 corresponds to the value obtained by adding 1.5 standard deviations to the mean of the distribution.

For each of the four perturbation measures, we identify sensitive positions if at least one mutation at that position has a score above the corresponding cutoff. In other words, sensitive positions for a certain measure are all the positions in the protein with one or more sensitive mutations. Since a position can be sensitive for each of the four measures, we define the minimum count as the number of measures for which a position needs to be sensitive in order to be considered a structurally sensitive position (SSP). The positions defined as structurally sensitive will serve as our predictions for functionally sensitive positions (FSPs) (Fig 7).

Given a distance threshold and a protein, predictions are thus made based on a cutoff vector and a minimum count. For example, considering the perturbation cutoff vector (1,1,1,1), and minimum count of 2, predictions include all positions in the protein that have at least one mutation with perturbation measure score one standard deviation above average, for at least two of the four measures.

We will also consider structurally robust positions (SRPs), as the complement of SSPs for certain parameters. That is, all positions not defined as structurally sensitive will be considered structurally robust.

## Assessing accuracy of predictions

In order to test the predictions obtained from the perturbation network data, we identified functionally sensitive positions from the deep mutational scanning data. To have a single functional value to define FSPs, we first attempted to standardize the data and look at positions with values above a certain cutoff. However, given the different data distributions of the proteins, this yielded vastly different percentages of FSPs. Two proteins had no positions with mean values one standard deviation above average, and when considering 0.5 standard deviations, percentages ranged from 17% to 38%. This made predictions hard to evaluate and highly dependent on each individual protein and its functional-change distribution.

Instead, we evaluated positions with lowest mean functional-change value using the 40-percentile, and compared these with predictions made from different cutoff vectors and minimum counts. That is, we consider FSPs to be the top 40% of positions with a stronger functional loss. Rounding down from the number of positions times 0.4, we obtain a mean functional percentage of 39.8%.

Given a perturbation cutoff vector and a minimum count, we get a set of predictions, positions likely to be functionally sensitive based on their perturbation networks, and compare them to 40% of positions with lowest mean functional values. We considered two measures to score these predictions: the *recall*, i.e. what percentage of FSPs we were able to predict, and the *precision*, i.e. what percentage of our predictions were functionally sensitive.

Given a set of predictions, or SSPs, and a set of FSPs, the intersection of them represents the true positives. The precision and recall scores can then be expressed as:

$$\text{Precision} = \frac{\text{True Positives}}{\text{SSPs}} = \frac{\text{True Positives}}{\text{True Positives} + \text{False Positives}}$$

$$\text{Recall} = \frac{\text{True Positives}}{\text{FSPs}} = \frac{\text{True Positives}}{\text{True Positives} + \text{False Negatives}}$$

The prediction percentage represents the ratio between SSPs and the total number of positions in the protein, while the functional percentage represents the ratio between FSPs and total positions:

$$\text{Prediction Percentage} = \frac{\text{SSPs}}{\text{Total Positions}} = \frac{\text{True Positives} + \text{False Positives}}{\text{Total Positions}}$$

$$\text{Functional Percentage} = \frac{\text{FSPs}}{\text{Total Positions}} = \frac{\text{True Positives} + \text{False Negatives}}{\text{Total Positions}}$$

We can also think about the precision and recall scores in terms of conditional probabilities:

$$\text{Precision} = \mathbb{P}[x \in \text{FSPs} \mid x \in \text{SSPs}] = \frac{\mathbb{P}[x \in \text{FSPs} \wedge x \in \text{SSPs}]}{\mathbb{P}[x \in \text{SSPs}]}$$

$$\text{Recall} = \mathbb{P}[x \in \text{SSPs} \mid x \in \text{FSPs}] = \frac{\mathbb{P}[x \in \text{FSPs} \wedge x \in \text{SSPs}]}{\mathbb{P}[x \in \text{FSPs}]}$$

Assuming independence between a position being functionally sensitive ($x \in \text{FSPs}$) and a position being structurally sensitive ($x \in \text{SSPs}$)—as would be the case if predictions were done randomly—we obtain that:

$$\text{Null Precision} = \frac{\mathbb{P}[x \in \text{FSPs} \wedge x \in \text{SSPs}]}{\mathbb{P}[x \in \text{SSPs}]} = \frac{\mathbb{P}[x \in \text{FSPs}] \cdot \mathbb{P}[x \in \text{SSPs}]}{\mathbb{P}[x \in \text{SSPs}]}$$
$$= \mathbb{P}[x \in \text{FSPs}] = \frac{\text{FSPs}}{\text{Total Positions}} = \text{Functional Percentage}$$

$$\text{Null Recall} = \frac{\mathbb{P}[x \in \text{SSPs} \wedge x \in \text{FSPs}]}{\mathbb{P}[x \in \text{FSPs}]} = \frac{\mathbb{P}[x \in \text{FSPs}] \cdot \mathbb{P}[x \in \text{SSPs}]}{\mathbb{P}[x \in \text{FSPs}]}$$
$$= \mathbb{P}[x \in \text{SSPs}] = \frac{\text{SSPs}}{\text{Total Positions}} = \text{Prediction Percentage}$$

Given these null scores, which would result from random predictions, we can obtain a single improvement score by dividing the corresponding real and null values:

$$\text{Improvement Score} = \frac{\text{Real Recall}}{\text{Null Recall}} = \frac{\text{Real Precision}}{\text{Null Precision}} = \frac{\text{True Positives} \times \text{Total Positions}}{\text{FSPs} \times \text{SSPs}}$$

## Comparison with state of the art methods

We compared our predictions with those obtained from DynaMut [73], which generates a prediction of the impact of a mutation on protein stability. Due to time constraints, and as this

software is implemented on a web server, we performed an alanine scan instead of obtaining all point mutations, with results taking around 3 and a half weeks. We considered the obtained value for $\Delta\Delta G$ from DynaMut for each position, and compared the mean value for our predictions maximizing accuracy, maximizing recall, and the positions not predicted to be functionally sensitive in either case. We found a general agreement with the results, as shown in S8 Fig, with both sets of predicted functionally sensitive positions consistently obtaining a lower $\Delta\Delta G$ than positions not predicted, indicating a mutation that makes the protein less stable. Considering that $\Delta\Delta G$ was calculated for a single mutation per position, compared to our thorough mutational scan, we believe that the results show a good agreement between the two approaches.

## Supporting information

**S1 Fig. Predictions based on individual measures, considering cutoff 1.5, comparing precision and recall scores obtained from varying the threshold for measures nodes, edges, weight, and diameter, from 3 Å to 10 Å.**
(TIF)

**S2 Fig. Precision and recall across 51 different perturbation cutoffs, ranging from 1 to 2 in intervals of 0.02.** Each row and column represents a different minimum count and protein, respectively.
(TIF)

**S3 Fig. Matrix of normalized structural change across all mutations for protein PSD95$^{pdz3}$.** Red and blue colors represent structural loss and robustness, respectively.
(TIF)

**S4 Fig. Matrix of normalized structural change across all mutations for protein PTEN.** Red and blue colors represent structural loss and robustness, respectively.
(TIF)

**S5 Fig. Matrix of normalized structural change across all mutations for protein APH(3')II.** Red and blue colors represent structural loss and robustness, respectively.
(TIF)

**S6 Fig. Matrix of normalized structural change across all mutations for protein SRC CD.** Red and blue colors represent structural loss and robustness, respectively.
(TIF)

**S7 Fig. Matrix of normalized structural change across all mutations for protein VIM-2.** Red and blue colors represent structural loss and robustness, respectively.
(TIF)

**S8 Fig. Point plot displaying the mean $\Delta\Delta G$ obtained from DynaMut [73] for three sets of positions for each of the five proteins studied, those included in the maximum precision prediction, those included in the maximum recall prediction, and those not included in either.**
(TIF)

## Acknowledgments

EHL is a recipient of the 2016 Marcos Moshinsky Fellowship in the Physical Sciences. CSV is an undergraduate student from the Program in Genomic Sciences, Universidad Nacional Autónoma de México (UNAM).

## Author Contributions

**Conceptualization:** Rodrigo Dorantes-Gilardi.

**Data curation:** Cristina Sotomayor-Vivas.

**Funding acquisition:** Enrique Hernández-Lemus.

**Methodology:** Cristina Sotomayor-Vivas, Rodrigo Dorantes-Gilardi.

**Project administration:** Enrique Hernández-Lemus.

**Resources:** Enrique Hernández-Lemus.

**Software:** Cristina Sotomayor-Vivas, Rodrigo Dorantes-Gilardi.

**Supervision:** Rodrigo Dorantes-Gilardi.

**Validation:** Cristina Sotomayor-Vivas, Enrique Hernández-Lemus, Rodrigo Dorantes-Gilardi.

**Visualization:** Cristina Sotomayor-Vivas, Rodrigo Dorantes-Gilardi.

**Writing – original draft:** Cristina Sotomayor-Vivas, Enrique Hernández-Lemus, Rodrigo Dorantes-Gilardi.

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
