## [Decision Letter · Decision Letter 0]

5 Oct 2021

PONE-D-21-15101Linking protein structural and functional change to mutation using amino acid networksPLOS ONE

Dear Dr. Dorantes-Gilardi,

Thank you for submitting your manuscript to PLOS ONE. After careful consideration, we feel that it has merit but does not fully meet PLOS ONE’s publication criteria as it currently stands. Therefore, we invite you to submit a revised version of the manuscript that addresses the points raised during the review process.

ACADEMIC EDITOR: Authors are requested to update the manuscript as per the suggestions by both the reviewers.

We look forward to receiving your revised manuscript.

Kind regards,

Sriparna Saha, PhD

Academic Editor

PLOS ONE

Journal Requirements:

“EHL is an Academic Editor at PLoS ONE.”

Reviewers' comments:

Reviewer's Responses to Questions

**Comments to the Author**

1. Is the manuscript technically sound, and do the data support the conclusions?

Reviewer #1: Yes

Reviewer #2: Partly

2. Has the statistical analysis been performed appropriately and rigorously? 

Reviewer #1: Yes

Reviewer #2: No

3. Have the authors made all data underlying the findings in their manuscript fully available?

Reviewer #1: Yes

Reviewer #2: Yes

4. Is the manuscript presented in an intelligible fashion and written in standard English?

Reviewer #1: Yes

Reviewer #2: Yes

5. Review Comments to the Author

Reviewer #1: The article is very interesting & informative. However I have few minor comments/queries/suggestions for betterment-

1. Line no. 77-89: Most of this part will go to methods section

2. Line no. 90-118- Most of this part has been discussed in the section of "results & discussion" so here it becomes redundant.

3. Line no. 92--- Which 5 protein & why these ? mention here itself.

4. Line no. 98. Write full form of ROC.

5. Line no. 101-102-- I think you have considered four different parameter (but here you have mentioned three)

6. Line no. 125-126: (In general ….. of the protein) The statement is not clear to me. If possible put a reference here (to support the statement)

7. I would recommend to summarize the major findings (with few bullets) at the beginning of the result & discussion section

8. Line no. 396-400- You may cite a more recent study here (Liu X, Luo Y, Li P, Song S, Peng J (2021) Deep geometric representations for modeling effects of mutations on protein-protein binding affinity. PLoS Comput Biol 17(8): e1009284. https://doi.org/10.1371/journal.pcbi.1009284)

9. You should write a small paragraph mentioning the strengths (e.g. inclusive design, statically methods etc.) & weakness (only considered perturbation data not deletion data) of your model/study

10. Lastly title & abbreviations may be incorporated in the figure ( in the picture itself) so that the figures can stand alone.

Reviewer #2: The article titled "Linking protein structural and functional change to mutation using amino acid networks" tries to establish a relation between structural and functional changes in terms of mutational dynamics of protein sequences. Authors use network modelling to study the relation between protein structure and functional variations considering 5 different proteins validating from deep mutational scanning databases. The idea of finding the structurally sensitive positions by observing its behaviour in the perturbation network seems interesting. Authors also claim that there is a strong correlation between the structurally sensitive positions (SSP) and functionally sensitive positions(FSP). They predicted FSPs from SSPs with a mean precision of 74.7% and recall of 69.3% for all the 5 proteins studied.

Though the article demonstrates various empirical results with statistical validations, the results are mainly some numerical values, in most of the cases, with no/inadequate explanation.

Some majors issues citing this observation are as follows:

1. Line 458, "We consider four topological attributes of P(t), namely its size (referred here as `nodes'), number of edges (`edges'), total sum of weights (`weight'), and its diameter (maximal smallest path, called here `diameter')."

It was never explained intuitively why these measures have been chosen. There could have been several other properties of a graph to choose from. During experimentation, a correlation has been established with these measures using a statistical count. Whereas, in a perturbation network, the no. of nodes indicates the number of positions affected or perturbed due to the mutation considered. The no. of edges indicate the connection between these nodes/positions affected. Similarly, the summation may capture the score of the total impact caused by this mutation. And, finally, the diameter encompasses the reach or spread of the perturbation in the network.

This kind of intuitive explanation could be helpful to the readers.

2. Again in line 154, "In the case of the measure 'diameter', correlations peaked between 3.5{3.8  A for all five proteins and then decreased for higher distance thresholds.", some measures have been reported with no proper explanation. Any intuition why the behaviour of diameter is so different from the others?

The authors have decided not to include diameter in the prediction model. However, diameter could have been an important measure to indicate the spread of the perturbation, which has been lost from sight because of some values. If it really is not that important for the prediction, then it should be justified with proper explanation.

3. Line 169, "We found that the number of nodes had the highest mean precision (72.66%), weight had the highest mean recall (71.76%), and diameter had the lowest score in both cases (52.58% and 49.02%, for precision and recall, respectively."

Again, there is an observation with inadequate explanation. Why is the result suggesting that this measure is crucial over others? No explanation!!

4. How are the standardized data arrays obtained in Figure 4? Why are the figures 4(A) and 4(B) looking different from C, D, and E. Are they representing the same color code? No explanation of the figures.

5. There should be a detailed stepwise explanation of Figure 7. First of all, is it "distance" or diameter in figure 7A? How are the positive and negative fraction values appearing in the tables of Figure 7B? If it is the output of standardization, then it should be properly explained. Maybe, the standardization was inappropriate which results in a differential behaviour of diameter in comparison to others.

Moreover, Standard Deviation encompasses the deviation (lower/higher) in the distribution. Does it mean that the smaller changes in the perturbation measures i.e., those which are less than the mean values should also be filtered out? Is the cutoff based on standard deviation meaningful here?

6. In line 493, authors say that "For each of the four perturbation measures, we identify sensitive positions if at least one mutation at that position has a score above the corresponding cutoff. In other words, sensitive positions for a certain measure are all the positions in the protein with one or more sensitive mutations."

Is it significant to call a position sensitive, if only 1 out of 10 mutations is showing a structural change in the perturbation network. Because 9 other mutations of the same position are demonstrating no structural change. Are all measures equally affected by a mutation at a particular position? If not, this needs to be studied methodically. Because, the authors already claimed that whether a mutation is effective in causing any functional change is dependent on the position of the mutation not on mutation itself.

This should be clarified with proper justification.

7. Any justification on deciding the cut-off vector to be [1.5,1.5,1.5,1.5]? In line 514, authors mentioned that "Two proteins had no positions with mean values one standard deviation above average, and when considering 0.5 standard deviations, percentages ranged from 17% to 38%." Then why the value 1.5?

8. The results presented in this article seem incomplete without the comparisons with some state of the art methods ( as mentioned in [1] and [2]) which predicted protein structure changes as result of mutational variation in the sequence.

[1] Carlos HM Rodrigues, Douglas EV Pires, David B Ascher, DynaMut: predicting the impact of mutations on protein conformation, flexibility and stability, Nucleic Acids Research, Volume 46, Issue W1, 2 July 2018, Pages W350–W355, https://doi.org/10.1093/nar/gky300

[2]Lijun Quan, Qiang Lv, Yang Zhang, STRUM: structure-based prediction of protein stability changes upon single-point mutation, Bioinformatics, Volume 32, Issue 19, 1 October 2016, Pages 2936–2946, https://doi.org/10.1093/bioinformatics/btw361

A demonstration of mutations and structural changes that are predicted by the proposed article, are also concurred by the methods like DynaMut and STRUM would be an interesting experimentation. If these two methods result in different findings than the proposed one, then this should be explained and validated as well.

6. PLOS authors have the option to publish the peer review history of their article (what does this mean?). If published, this will include your full peer review and any attached files.

Reviewer #1: **Yes: **Abhijit Dey

Reviewer #2: **Yes: **Angana Chakraborty, PhD

---

## [Author Response · Author response to Decision Letter 0]

23 Nov 2021

Dear Dr. Sriparna Saha, Academic Editor:

We would like to take the opportunity to thank you as well as the reviewers for your valuable feedback. We are sure our manuscript has considerably been improved as a consequence.

Please find below the response to each point raised by the reviewers. When specific lines in the manuscript are mentioned we refer to the revised manuscript with track changes.

Reviewer #1:

The article is very interesting & informative. However I have few minor comments/queries/suggestions for betterment-

The authors would like to thank Reviewer 1 for their professional critique and assessment of our work. In what follows we will present a point-by-point response to their comments and suggestions.

1. Line no. 77-89: Most of this part will go to methods section

We have revised our manuscript accordingly with your suggestions in lines 81--90 of the revised manuscript with track changes. Most of the paragraph was taken to the methods of the manuscript.

2. Line no. 90-118- Most of this part has been discussed in the section of "results & discussion" so here it becomes redundant.

This section has been re-written to eliminate redundancies.

3. Line no. 92--- Which 5 protein & why these ? mention here itself.

We selected these proteins based on the completeness of the point mutations evaluated, as well as a focus on experimental assays evaluating enzyme binding. We also considered that the size of the proteins was not too large, as all point mutations were evaluated. The manuscript has been updated to include that at this point in the introduction and methodology. 

4. Line no. 98. Write full form of ROC.

The manuscript has been updated with its full form. 

5. Line no. 101-102-- I think you have considered four different parameter (but here you have mentioned three)

As stated in the methods section, the parameter for diameter was ultimately not considered for predictions, and because of that it is not mentioned here. We have updated lines 182--185 of the manuscript with track changes to include an explanation of its exclusion. 

6. Line no. 125-126: (In general ….. of the protein) The statement is not clear to me. If possible put a reference here (to support the statement)

We have rephrased this section and included some supporting references in lines 132--134.

7. I would recommend to summarize the major findings (with few bullets) at the beginning of the result & discussion section

A summary of the major findings was included at the beginning of the section Results and discussion lines 135--140 of the revised manuscript with tracked changes.

8. Line no. 396-400- You may cite a more recent study here (Liu X, Luo Y, Li P, Song S, Peng J (2021) Deep geometric representations for modeling effects of mutations on protein-protein binding affinity. PLoS Comput Biol 17(8): e1009284. https://doi.org/10.1371/journal.pcbi.1009284)

We have added this reference to the concluding remarks of the manuscript in line 406.

9. You should write a small paragraph mentioning the strengths (e.g. inclusive design, statically methods etc.) & weakness (only considered perturbation data not deletion data) of your model/study

A brief subsection on the scope and limitations of our work has been included in the revised version of our manuscript in lines 434--440.

10. Lastly title & abbreviations may be incorporated in the figure ( in the picture itself) so that the figures can stand alone.

Fig 4, 6, and 7 were updated to include additional information including titles and abbreviations.

Reviewer #2: 

The article titled "Linking protein structural and functional change to mutation using amino acid networks" tries to establish a relation between structural and functional changes in terms of mutational dynamics of protein sequences. Authors use network modelling to study the relation between protein structure and functional variations considering 5 different proteins validating from deep mutational scanning databases. The idea of finding the structurally sensitive positions by observing its behaviour in the perturbation network seems interesting. Authors also claim that there is a strong correlation between the structurally sensitive positions (SSP) and functionally sensitive positions(FSP). They predicted FSPs from SSPs with a mean precision of 74.7% and recall of 69.3% for all the 5 proteins studied.

Though the article demonstrates various empirical results with statistical validations, the results are mainly some numerical values, in most of the cases, with no/inadequate explanation.

The authors are thankful to Reviewer 2 for their professional academic review of our manuscript. Below we will present a point-by-point response to your comments and concerns.

Some majors issues citing this observation are as follows:

1. Line 458, "We consider four topological attributes of P(t), namely its size (referred here as `nodes'), number of edges (`edges'), total sum of weights (`weight'), and its diameter (maximal smallest path, called here `diameter')."

It was never explained intuitively why these measures have been chosen. There could have been several other properties of a graph to choose from. During experimentation, a correlation has been established with these measures using a statistical count. Whereas, in a perturbation network, the no. of nodes indicates the number of positions affected or perturbed due to the mutation considered. The no. of edges indicate the connection between these nodes/positions affected. Similarly, the summation may capture the score of the total impact caused by this mutation. And, finally, the diameter encompasses the reach or spread of the perturbation in the network.

This kind of intuitive explanation could be helpful to the readers.

The manuscript has been revised to present a clearer image of the reasons to use such topological attributes, as well as the rationale behind the statistical analysis and the scoring procedures. This can be seen in the (new) lines 104 to 107 in the revised manuscript (see the manuscript with tracked changes for easy reading). 

2. Again in line 154, "In the case of the measure 'diameter', correlations peaked between 3.5{3.8  A for all five proteins and then decreased for higher distance thresholds.", some measures have been reported with no proper explanation. Any intuition why the behaviour of diameter is so different from the others?

The authors have decided not to include diameter in the prediction model. However, diameter could have been an important measure to indicate the spread of the perturbation, which has been lost from sight because of some values. If it really is not that important for the prediction, then it should be justified with proper explanation.

Reviewer 2 is right in that our former presentation was not clear enough on these issues, for this reason we have modified the manuscript to make this more evident (lines 181-184). Diameter, in particular, was not included because we believe that it was too sensitive to small changes in thresholds, as can be seen in Figure 1, in addition to the low recall and precision scores obtained from predictions based on diameter alone. We believe that this measure is too sensitive as adding or removing a single edge could significantly change the maximal smallest path without significantly changing the network itself. 

3. Line 169, "We found that the number of nodes had the highest mean precision (72.66%), weight had the highest mean recall (71.76%), and diameter had the lowest score in both cases (52.58% and 49.02%, for precision and recall, respectively."

Again, there is an observation with inadequate explanation. Why is the result suggesting that this measure is crucial over others? No explanation!!

We believe that the difference between the diameter and the other measures lies on its sensitivity to mutations: Given that the perturbation network is typically small, a loss of an edge could separate the network such that the position being mutation remains in an even smaller connected component, giving a very small diameter. This is reflected in its poor performance when taken individually to predict functional perturbation (a difference of 20% with precision and recall of other measures). However, the inclusion of the number of edges can serve as a proxy to the extension of the network. An explicit mention of the exclusion of the diameter was added in the manuscript.

4. How are the standardized data arrays obtained in Figure 4? Why are the figures 4(A) and 4(B) looking different from C, D, and E. Are they representing the same color code? No explanation of the figures.

We have updated the manuscript to clarify the standardization in a subsection of the methodology, which was done by subtracting the mean and dividing by the standard deviation. Figures 4A and 4B show experimental and computational data for each point mutation, respectively, while Figures C, D and E show which positions were predicted in each case, and the mean functional value from the experimental data. The manuscript and figure caption has been updated to clarify this. 

The explanation of the color code for each plot was included as requested by the reviewer.

5. There should be a detailed stepwise explanation of Figure 7. First of all, is it "distance" or diameter in figure 7A? How are the positive and negative fraction values appearing in the tables of Figure 7B? If it is the output of standardization, then it should be properly explained. Maybe, the standardization was inappropriate which results in a differential behaviour of diameter in comparison to others.

Figure 7 has been updated to include the correct term (diameter) and the origin of the values.

Moreover, Standard Deviation encompasses the deviation (lower/higher) in the distribution. Does it mean that the smaller changes in the perturbation measures i.e., those which are less than the mean values should also be filtered out? Is the cutoff based on standard deviation meaningful here?

As we are considering the perturbation network as the absolute value of the difference between the two networks (mutation network and wild type network), for all measures, a value of zero would be expected if the networks were identical, and no negative values are possible. Therefore, values below the mean are closer to the original network, and we only filter values above the mean. The manuscript has been updated to better reflect this in the methods section. 

6. In line 493, authors say that "For each of the four perturbation measures, we identify sensitive positions if at least one mutation at that position has a score above the corresponding cutoff. In other words, sensitive positions for a certain measure are all the positions in the protein with one or more sensitive mutations."

Is it significant to call a position sensitive, if only 1 out of 10 mutations is showing a structural change in the perturbation network? Because 9 other mutations of the same position are demonstrating no structural change. Are all measures equally affected by a mutation at a particular position? If not, this needs to be studied methodically. Because, the authors already claimed that whether a mutation is effective in causing any functional change is dependent on the position of the mutation not on mutation itself.

This should be clarified with proper justification.

This is a valid point raised by the reviewer. As it is true that structural perturbation is based on the position being mutated instead of the particular mutation, standardization of the data is made considering the full set of mutations and their effect on the structure of the original protein. In general, the presence of a mutation that alters the structure in such a way suggests that the other mutations also alter the structure although probably at weaker levels. This is shown in the manuscript in Table 2 by taking the percentage of mutations with the same positive or negative symbol for each position. 

We are working with computationally obtained approximations of what the structure might look like if the protein was mutated. Based on this idea, we believe that in vivo mutations could distort the structure more than what is shown in in silico mutations, and therefore considered that a single “bad” mutation was a good indication of the possibility of a sensitive neighborhood in the protein. In other words, we believe that it is more likely that mutations that appear to not alter the structure have a bigger effect in vivo than viceversa. We have updated the manuscript to better reflect this reasoning. 

7. Any justification on deciding the cut-off vector to be [1.5,1.5,1.5,1.5]? In line 514, authors mentioned that "Two proteins had no positions with mean values one standard deviation above average, and when considering 0.5 standard deviations, percentages ranged from 17% to 38%." Then why the value 1.5?

The quote refers to the experimental data we used, which we found to have varying distributions and few positions above the standard deviation cutoffs. This led us to filter experimental data by quantiles, settling on 40%. On the other hand, the computational data obtained from in silico mutations had much more even distributions, so using standard deviation cutoffs was possible. The value 1.5 was selected based on how many positions passed the cutoff. The manuscript has been updated to better explain the distinction between the filtering of the data in the “data standardization” subsection of the methodology.

8. The results presented in this article seem incomplete without the comparisons with some state of the art methods ( as mentioned in [1] and [2]) which predicted protein structure changes as result of mutational variation in the sequence.

[1] Carlos HM Rodrigues, Douglas EV Pires, David B Ascher, DynaMut: predicting the impact of mutations on protein conformation, flexibility and stability, Nucleic Acids Research, Volume 46, Issue W1, 2 July 2018, Pages W350–W355, https://doi.org/10.1093/nar/gky300

[2]Lijun Quan, Qiang Lv, Yang Zhang, STRUM: structure-based prediction of protein stability changes upon single-point mutation, Bioinformatics, Volume 32, Issue 19, 1 October 2016, Pages 2936–2946, https://doi.org/10.1093/bioinformatics/btw361

A demonstration of mutations and structural changes that are predicted by the proposed article, are also concurred by the methods like DynaMut and STRUM would be an interesting experimentation. If these two methods result in different findings than the proposed one, then this should be explained and validated as well.

We submitted the proteins studied to both the DynaMut and STRUM web servers by October 19th, however, we only received the results from DynaMut. We found general agreement with our results which we discuss in a new methodology section of the manuscript. We especially appreciate this comment as we believe it brought an important addition to our manuscript.

---

## [Decision Letter · Decision Letter 1]

13 Dec 2021

Linking protein structural and functional change to mutation using amino acid networks

PONE-D-21-15101R1

Dear Dr. Dorantes-Gilardi,

We’re pleased to inform you that your manuscript has been judged scientifically suitable for publication and will be formally accepted for publication once it meets all outstanding technical requirements.

Kind regards,

Sriparna Saha, PhD

Academic Editor

PLOS ONE

Additional Editor Comments (optional):

Reviewers' comments:

Reviewer's Responses to Questions

**Comments to the Author**

1. If the authors have adequately addressed your comments raised in a previous round of review and you feel that this manuscript is now acceptable for publication, you may indicate that here to bypass the “Comments to the Author” section, enter your conflict of interest statement in the “Confidential to Editor” section, and submit your "Accept" recommendation.

Reviewer #1: All comments have been addressed

Reviewer #2: All comments have been addressed

2. Is the manuscript technically sound, and do the data support the conclusions?

Reviewer #1: Yes

Reviewer #2: Yes

3. Has the statistical analysis been performed appropriately and rigorously? 

Reviewer #1: Yes

Reviewer #2: Yes

4. Have the authors made all data underlying the findings in their manuscript fully available?

Reviewer #1: Yes

Reviewer #2: Yes

5. Is the manuscript presented in an intelligible fashion and written in standard English?

Reviewer #1: Yes

Reviewer #2: Yes

6. Review Comments to the Author

Reviewer #1: The authors have addressed the quires adequately. Now the manuscripts is looking much better. The editor may accept it for Publication.

Reviewer #2: (No Response)

7. PLOS authors have the option to publish the peer review history of their article (what does this mean?). If published, this will include your full peer review and any attached files.

Reviewer #1: **Yes: **Abhijit Dey

Reviewer #2: **Yes: **Dr. Angana Chakraborty

---

## [Editor Report · Acceptance letter]

17 Dec 2021

PONE-D-21-15101R1 

Linking protein structural and functional change to mutation using amino acid networks 

Dear Dr. Dorantes-Gilardi:

I'm pleased to inform you that your manuscript has been deemed suitable for publication in PLOS ONE. Congratulations! Your manuscript is now with our production department. 

Kind regards, 

on behalf of

Dr. Sriparna Saha 

Academic Editor

PLOS ONE